# Effect of caffeine on neuromuscular function following eccentric-based exercise

Ana C. Santos-Mariano[1,2,3], Fabiano Tomazini[1,2,3], Leandro C. Felippe[1], Daniel Boari[4], Romulo Bertuzzi[5], Fernando R. De-Oliveira[6], Adriano E. Lima-Silva[1,2,3]*

1 Sport Science Research Group, Academic Center of Vitoria, Federal University of Pernambuco, Vitoria de Santo Antao, Pernambuco, Brazil, 2 Human Performance Research Group, Academic Department of Physical Education, Federal University of Technology – Parana, Curitiba, Parana, Brazil, 3 Human Performance Research Group, Federal University of Parana, Curitiba, Parana, Brazil, 4 Center of Engineering, Modeling, and Applied Social Sciences, Federal University of ABC, Sao Bernardo do Campo, Sao Paulo, Brazil, 5 Endurance Performance Research (GEDAE-USP), School of Sport and Physical Education and Sport, University of Sao Paulo, Sao Paulo, Brazil, 6 Center for Studies of Human Movement, Department of Physical Education, Federal University of Lavras, Minas Gerais, Brazil

* aesilva@utfpr.edu.br

**Data Availability Statement:** All data files presented in this study are available in the database Open Science Framework (https://osf.io), DOI: 10. 17605/OSF.IO/H6MN8.

## Abstract

This study investigated the effect of caffeine on neuromuscular function, power and sprint performance during the days following an eccentric-based exercise. Using a randomly counterbalanced, crossover and double-blinded design, eleven male jumpers and sprinters (age: 18.7 ± 2.7 years) performed a half-squat exercise (4 x 12 repetitions at 70% of 1 RM), with eccentric action emphasized by using a flexible strip attached to their knees (*Tirante Musculador®*). They ingested either a capsule of placebo or caffeine (5 mg.kg$^{-1}$ body mass) 24, 48 and 72 h after. Neuromuscular function and muscle power (vertical countermovement-jump test) were assessed before and after the half-squat exercise and 50 min after the placebo or caffeine ingestion at each time-point post-exercise. Sprint performance was measured at pre-test and 75 min after the placebo or caffeine ingestion at each time-point post-exercise. Maximal voluntary contraction (overall fatigue) and twitch torque (peripheral fatigue) reduced after the half-squat exercise (-11 and -28%, respectively, P < 0.05) but returned to baseline 24 h post-exercise (P > 0.05) and were not affected by caffeine ingestion (P > 0.05). The voluntary activation (central fatigue) and sprint performance were not altered throughout the experiment and were not different between caffeine and placebo. However, caffeine increased height and power during the vertical countermovement-jump test at 48 and 72 h post half-squat exercise, when compared to the placebo (P < 0.05). In conclusion, caffeine improves muscle power 48 and 72 h after an eccentric-based exercise, but it has no effect on neuromuscular function and sprint performance.

## Introduction

Strength training with eccentric emphasis is used by sprinters and jumpers to provoke in-series sarcomere hypertrophy, which could increase the velocity of muscle contraction [1].

**Funding:** We declare this study was financed in part by the Coordenação de Aperfeiçoamento de Pessoal de Nível Superior (CAPES) - Brazil (http://www.capes.gov.br/) - Finance Code "001". Fabiano Tomazini and Ana C. Santos-Mariano were also awarded with a PhD scholarship from CAPES. The authors received no specific funding for this work. We also declare there was no additional external funding received for this study.

**Competing interests:** The authors have declared that no competing interests exist.

However, exercise with eccentric emphasis also increases muscle damage via increased sarcoplasmic rupture, dilation of the transverse tubule system, myofibrillar component distortion and sarcoplasmic reticulum fragmentation [2,3]. Due these structural changes, there is a late increase in creatine kinase (CK), a marker of muscle damage, with a concomitant increase in muscle pain after an eccentric exercise (24 to 72h post-exercise), the so-called delayed onset muscle soreness (DOMS). Increased muscle damage and DOMS might carry over into exacerbated neuromuscular fatigue, which can reduce performance during the subsequent training sessions even in individuals accustomed to this mode of training [4–6].

Only a few studies have investigated the time course of neuromuscular fatigue after eccentric exercise [4–8]. While it has consistently been demonstrated that DOMS, measured with a 10-point Pain Intensity Scale [9], peaks 48 h after an eccentric exercise, the time course of neuromuscular fatigue after an eccentric exercise is more controversial. Maximal voluntary contraction (MVC), an indicator of overall fatigue, can remain reduced from 24 h to 8 days or more after an eccentric exercise [4,8]. Central fatigue, measured by voluntary activation (VA) via the twitch interpolated technique is restored within 24 h to 72 h [4,6,10], while twitch torque (an indicator of peripheral fatigue) is restored in 24 h [10], 48 h [8] or sometimes more than 4 days [4,6]. Part of these inconsistencies might be due to different protocols inducing muscle damage such as elbow flexion [6], backward downhill walking [10] or heavy-resistance (strength) and jump training [8]. It is noteworthy that only one study recruited athletes as volunteers and used exercises that are part of the training routine of sprinters and jumpers [8]. Therefore, the time course of central and peripheral fatigue after an exercise with eccentric emphasis typically used in athlete's training routine is underexplored in well-trained athletes and deserves further investigation.

Because athletes train almost every day, it is important to maintain performance during subsequent training sessions [11]. In this sense, caffeine (1,3,7-trimethylxanthine) seems to be a promising alternative. Following its removal from the World Anti-Doping Agency (WADA) Prohibited List in 2004, the use of caffeine for track and field athletes increased substantially [12]. The use of caffeine has been reported to be large in national/international level athletes (~48%) and in power athletes (~57%) (12). Because caffeine acts on the central nervous system (CNS) as an adenosine receptor antagonist, caffeine maintains neuronal excitability and counteracts central fatigue. Caffeine has also analgesic properties that could reduce the perception of pain during exercise [13]. In addition, caffeine improves excitation-contraction coupling by increasing calcium release/reuptake and $Na^+$-$K^{+-}$ATPase pump activity [14]. Caffeine also increases peak isokinetic torque after activities that resulted in exercise-induced muscle damage [15]. Considering these effects, caffeine might be an effective alternative to counteract the negative effect of DOMS provoked by a prior exercise with eccentric emphasis on power, sprint performance and central and peripheral fatigue.

Therefore, the first aim of the present study was to investigate the time course of CK, DOMS, jump and sprint performance and central and peripheral components of neuromuscular fatigue after a half-squat exercise with eccentric emphasis. The second aim was to investigate whether caffeine would influence central and peripheral components of neuromuscular fatigue and jump and sprint performance during the following days after the half-squat exercise with eccentric emphasis. We hypothesized that acute caffeine ingestion would improve central and peripheral components of neuromuscular fatigue and jump and sprint performance in the following days after a half-squat exercise with eccentric emphasis.

## Materials and methods

### Participants

Eleven well-trained young males' sprinters and jumpers (age: 18.7 ± 2.7 years old; weight: 69.9 ± 6.4 kg; height: 180.7 ± 7.7 cm), accustomed to eccentric training and with low-to-moderate habitual caffeine consumption ($< 80$ mg.day$^{-1}$) [16] participated in this study. The inclusion criteria were: (a) to be free from neuromuscular and musculoskeletal disorders; (b) to have had at least one year of experience with athletic training; (c) to have been training 5 to 6 times per week (12 to 15 hours per week) and; (d) to have been involved in national or international competitions in the year prior to the study. The participants and their guardians were informed about the procedures prior to the beginning of the study and signed a written informed consent. The research was conducted according to the Declaration of Helsinki and was approved by the Research Ethics Committee of the Federal University of Pernambuco.

### Experimental design

The study was carried out within a 3-week period of a preparatory phase of an annual training periodization. In this phase, training sessions were designed to improve general resistance and not focused in eccentric training. Two preliminary sessions were designed to assess one-repetition maximal strength in a half-squat exercise (1RM) using a *Tirante Musculador*® (see one-repetition maximal strength test section for details) and to familiarize the participants with the assessment of neuromuscular function. In the first preliminary session, anthropometric measurements were taken, and a full familiarization with the assessment of neuromuscular function and 1RM was performed. Forty-eight hours later, in the second preliminary session, 1RM was determined and participants had another opportunity to be familiarized with the assessment of neuromuscular function.

After two days of low-intensity training, premeasurements were performed over the next two days. On day 1, after a 20 min warm-up (10 minutes running + 10 minutes of dynamic stretching), pre-test sprints (one sprint of 30 m following by 5 min rest, one sprint of 50 m following by 7 min rest and one sprint of 100 m) were performed in an outdoor athletic track (Fig 1). On Day 2, baseline blood sample, DOMS, perceived recovery, neuromuscular function and vertical countermovement-jump test (CMJ) were assessed. Neuromuscular function was preceded by a warm-up (see neuromuscular function assessment section for details). Five minutes later, 4 sets of 12 repetitions at 70% of 1RM (2-min rest between sets) of a half-squat exercise with emphasis on eccentric action were performed using the *Tirante Musculador*®. After exercise, the neuromuscular function and CMJ performance were reassessed. During the next three days (days 3, 4 and 5), a blood sample was collected and participants ingested a gelatin capsule containing either 5 mg.kg$^{-1}$ body mass of anhydrous caffeine (CAF) or cellulose (PLA) 50 min before the assessment of DOMS, perceived recovery, neuromuscular function and CMJ. Participants also performed a sprint training session 75 min after the supplement ingestion.

After a 7-day washout period performing a low-intensity training program, participants repeated the procedures described above, but those who had ingested caffeine in the first moment ingested a placebo and those who had ingested a placebo ingested caffeine (crossover design). The order of CAF and PLA ingestion were randomly counterbalanced and supplements offered in a double-blind manner. All tests were performed at the same time of day to avoid any effect of the circadian variation. Participants refrained from alcohol and caffeinated beverages throughout the study. Participants were also instructed to record their food and drink intake starting 48 h before the Day 1 until the Day 5 and then to replicate this intake in

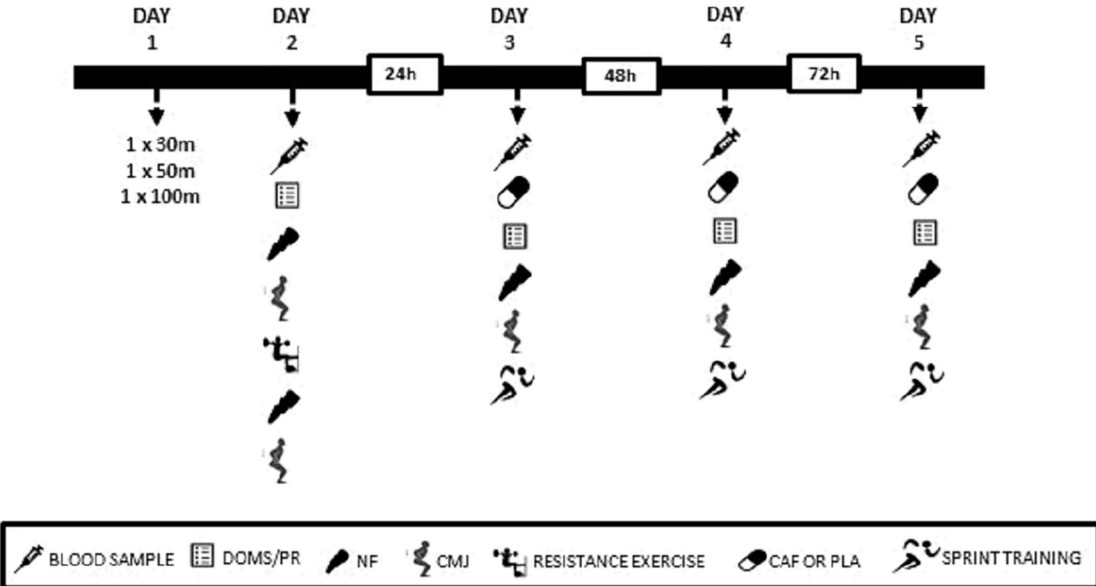

**Fig 1. Study timeline.** DOMS, Delayed Onset Muscle Soreness; PR, Perceived Recovery; NF, neuromuscular function; CMJ, countermovement jump; CAF, caffeine; PLA, placebo.

the second phase of the experiment. Participants were asked not to eat anything two hours before each test session.

## One-repetition maximal strength test

The 1RM test was performed during a half-squat exercise with participants coupled to a *Tirante Musculador*® (TMR-World, Barcelona, Spain). This accessory is a simple belt that allows anchor the calf and leave the body hanging with the gravity center far from the fulcrum of the knee and thus increasing the eccentric overload (Fig 2). The two flexible strips of the belt were attached below participants' knees and fixed to a column. Participants performed the flexion and extension of the knees (90˚ amplitude). This flexible strip has been demonstrated suitable to increase eccentric action during a half-squat exercise [17]. Participants were familiar with this accessory as they had used it in their training routine as a form to increase eccentric action without using an excessive external load, which may reduce the risk of injuries associated with eccentric-based training, mainly in a preparatory phase of an annual training periodization [17]. The participants performed a warm up consisting of 15 repetitions at ~ 50% of the estimated 1RM [18]. After that, participants rested for 2 min and the load was increased to the expected 1 RM. Five attempts were necessaries to achieve the 1RM load (i.e., the maximum load that could be lifted once using the proper technique), with a 2 min interval between the attempts, as previously suggested [18]. The 1RM load was 130.2 ± 18.6 kg.

## Half-squat exercise with eccentric emphasis

Participants performed 15 repetitions of half-squat exercise at 50% of 1RM as warm up and after a 2 min rest, performed 4 sets of 12 repetitions at 70% of 1RM, with 2 min rest between sets. Each repetition took 3 s (2 seconds for the eccentric phase and 1 second for the concentric phase) and was controlled using a metronome. The eccentric action was emphasized by participants performing the half-squat using the *Tirante Musculador*®, as detailed above. We ascertain this protocol with athetes's coach to simulate their training routine. We chose a protocol

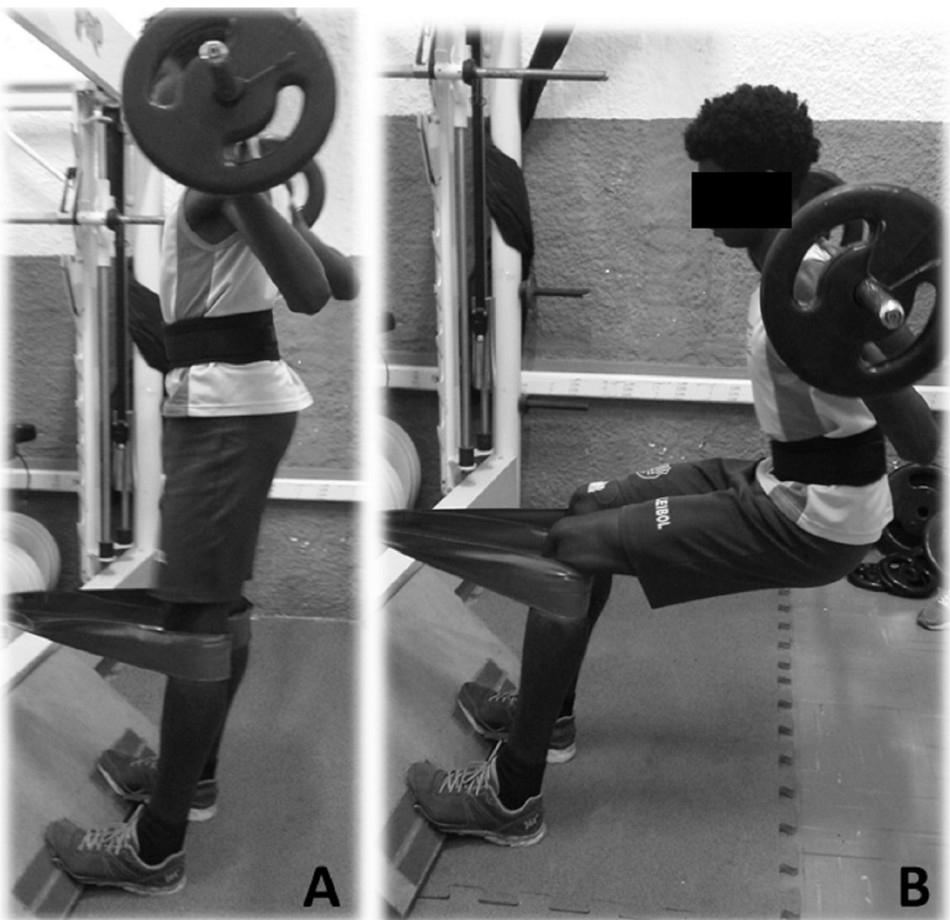

**Fig 2. Half-squat exercise with eccentric emphasis performed with the *Tirante Musculador*®.** A: Initial position of the movement; B: Final position of the movement.

able to provoke moderate rather than a more intense DOMS because athletes were in a preparatory phase of an annual training periodization; therefore, protocol to induced more intense DOMS would make athletes more susceptive to muscle injuries. The load at 70% of 1RM was 91.1 ± 13.9 kg.

## Neuromuscular function assessment

The neuromuscular function was assessed with the participants seated on a custom-made bench with their hip and knees angles set at 120˚ and 90˚, respectively. A non-compliant cuff attached to a calibrated linear strain gauge was fixed to the right ankle superior to the malleoli for force measurement (EMG System of Brazil, São Jose dos Campos, Brazil). A monopolar 0.5 cm diameter cathode electrode was positioned at the right femoral nerve for electrical stimulation (Ambu® Neuroline 715, Ballerup, Denmark). The anode was positioned on the gluteal fold opposite to the cathode. The position of the electrodes was marked with indelible ink to ensure identical placement in subsequent visits.

The optimal intensity of stimulation was determined by percutaneous electrical nerve stimuli (1 Hz and 80 μs duration) applied to the femoral nerve using an electrical stimulator (Neuro-TES, Neurosoft, Ivanovo, Russia). The intensity of stimulation began with 100 volts and increased 30 volts every 30 s until the occurrence of a plateau in the muscle membrane

excitability ($M_{wave}$) and quadriceps twitch torque ($Q_{tw}$). To ensure a supramaximal stimulus, the intensity of stimulation that $M_{wave}$ and $Q_{tw}$ plateaued was further increased by 20%. The optimal intensity of stimulation was determined in the first visit and checked before every subsequent visit ($301 \pm 49$ volts).

The neuromuscular function assessment (except immediately after the half-squat exercise) was preceded by a warm up composed of a 5 min jogging and 4 x 5 s isometric contractions of knee extension at ~60% of maximal subjective isometric force (30 s rest between contractions). The protocol of neuromuscular function assessment consisted of 6 x 5 s MVC of knee extensors, interspaced by 60-s recovery, as previously suggested [19]. Stimulus of 1 Hz (80 µs) was applied during each MVC (superimposed twitch) at the plateau of isometric force. Potentiated quadriceps twitch torque evoked by 1 Hz ($Q_{twpot}$) and paired pulses at 10 Hz ($Q_{tw10}$) and 100 Hz ($Q_{tw100}$) was measured 2, 4 and 6 s after each MVC, respectively. The $M_{wave}$ peak-to-peak amplitude was calculated for each 1 Hz stimulus. A $Q_{tw10} \cdot Q_{tw100}^{-1}$ ratio was calculated to identify low-frequency fatigue [20]. The VA was measured by the following equation [21]:

$$\%VA = (1 - \mathrm{sup}erimposed \div Qtwpot) \times 100 \qquad (1)$$

where superimposed is the difference between the force before the stimulus and the peak of force induced by the stimulus.

The average of the last four MVC for each time point was used for further statistical analysis of the neuromuscular function [19].

## Countermovement jump test

Participants were familiar with CMJ in their training routine. The participants started in a standing position, dropped to a squatting position and jump upwards as high as possible. The jumps were performed with hands on the hip. The flying time during the CMJ was measured using a contact mat (Jump System Pro, Cefise, São Paulo, Brazil). The CMJ height was calculated from flying time, while CMJ power calculated from flying time and body mass, using a commercial software (Jump System 1.0, Cefise, São Paulo, Brazil). The jumps were performed three times (10 s interval) and the mean height and power used for further statistical analysis.

## Sprint training

Sprint training was composed by a series of sprints similar to that performed in training routine. The sprint training session started with a 20 min warm-up (10 minutes running + 10 minutes of dynamic stretching). Then, participants performed: 1) 3 sets of 30-m sprints, with a 5 min rest between sets; 2) 3 sets of 50-m sprints, with a 7 min rest between sets and; 3) 3 sets of 100-m sprints, with a 10 min rest between sets. They were instructed to run as fast they could and wore the same footwear. Participants performed 10 min of running and stretching after training to cool down. The timing of the sprints was monitored using photocells (TC-System, Brower Timing System, US). Participants positioned immediately before the first photocell to start. The mean velocity in each distance was calculated and used for further analysis.

## Blood sample

Blood samples (8 ml) were drawn by a professional phlebotomist from the antecubital vein by venipuncture and transferred to tubes containing Clot activator and gel for serum separation (SST II Plus, BD Vacutainer®, USA). The serum CK concentration was determined by an enzymatic method using commercial kits (Labtest Diagnostica S.A., Minas Gerais, Brazil), with the resultant reaction reading in a spectrophotometer (Genesys 10 S UV-vis, Thermo Electron Scientific Instruments, Madison, WI, United States).

## Delayed onset muscle soreness and perceived recovery

The DOMS was measured with the Pain Intensity Scale ranging from 0 (no pain) to 10 (very intense pain, almost unbearable) [9]. The perception of recovery was measured on the Total Quality of Recovery Scale ranging from 6 to 20, where 6 is "nothing recovered" and 20 is "fully recovered" [22]. Participants were familiar with these scales as they had used them during their training routine.

## Statistical analyses

Because of the lack of data regarding the caffeine effects on neuromuscular function post an eccentric-based exercise in athletes, the required sample size was estimated using an effect size (ES = 0.37) reported in a meta-analysis investigating the effect of caffeine ingestion on muscular strength [23]. With an alpha of 0.05 and a desired power of 0.80, the total sample size necessary to achieve statistical significance was estimated to be 10 participants. However, the starting sample size was increased to 11 participants, assuming that 10% might drop out during the data collection. The sample size calculation was performed using G*Power software (version 3.1.9.2, Kiel University, Germany).

The Shapiro-Wilk test was performed to determine the normality of the data. The neuromuscular function parameters (MVC, VA, $M_{wave\ ampl}$, $Q_{twpot}$, $Q_{tw10}$, $Q_{tw100}$ and $Q_{tw10} \cdot Q_{tw100}^{-1}$), CMJ, CK and perceived recovery were analyzed using a two-way, repeated-measures ANOVA [supplement (PLA and CAF) x time (baseline, post-exercise and 24, 48 and 72 h post exercise)], with a Duncan post-hoc test being utilized when ANOVA detected significant main effects and/or interaction. The pre-test sprint performance before the first and second experimental blocks were compared using a paired $t$ test in order to check if athletes had fully recovered from the previous tests and training sessions. The performance during the sprint training was then analyzed separately using a two-way, repeated-measures ANOVA [supplement (PLA and CAF) x time (24, 48 and 72 h post exercise). The DOMS was analyzed using Friedman ANOVA followed by the Wilcoxon test because the DOMS had not shown a normal distribution. Partial eta squared ($\eta_p^2$) was also calculated as a measure of the effect size and classified as small ($\eta_p^2 < .06$), moderate ($.06 \leq \eta_p^2 < .15$) or large ($\eta_p^2 \geq .15$). The statistical significance was accepted at $P < 0.05$. Data are reported as mean ± SD, unless otherwise stated. Statistical analyses were performed using statistics package for Windows (version 10, StatSoft, Tulsa, OK, USA).

## Results

### Muscle damage, DOMS and perceived recovery

The CK increased above baseline levels 48 and 72 h post-exercise (main effect of time, $F_{(4,32)} = 10.810$, $P = 0.001$, $\eta_p^2 = 0.57$, Table 1). The DOMS increased from baseline to 24 h post-exercise, remaining above the baseline values until 72 h post-exercise ($\chi^2_{(10)} = 25.312$, $P = 0.001$). Compared to baseline, the perceived recovery was always incomplete at 24, 48 and 72 h post-exercise (main effect of time, $F_{(3,\ 27)} = 12.429$, $P = 0.001$, $\eta_p^2 = 0.58$, Table 1). There was no effect of supplement for CK, DOMS and perceived recovery ($P > 0.05$).

### Neuromuscular function

The MVC decreased significantly from baseline to immediate post-exercise (-11%) but recovered fully 24 h later (main effect of time, $F_{(4,32)} = 6.560$, $P = 0.001$, $\eta_p^2 = 0.45$, Fig 3A). Similarly, markers of peripheral fatigue ($Q_{twpot}$, $Q_{tw10}$ and $Q_{tw100}$) decreased significantly with the exercise (-30%, -35% and -12%, respectively) and recovered fully 24 h later (main effect of time,

**Table 1. Creatine kinase, delayed onset muscle soreness and perceived recovery at baseline, and at 24, 48 and 72 h later a half-squat exercise with placebo or caffeine ingestion.**

| | PLACEBO | | | | CAFFEINE | | | | | | |
|---|---|---|---|---|---|---|---|---|---|---|---|
| | Baseline | 24 h | 48 h | 72 h | Baseline | 24 h | 48 h | 72 h | $F$ | p | $\eta_p^2$ |
| **CK** (U/L) | 356.6 ± 220.1 | 381.9 ± 136.3 | 481.4 ± 66.3* | 491.6 ± 209.8* | 320.5 ± 148.5 | 337.9 ± 114.1 | 496.8 ±173.0* | 497.3 ± 163.6* | 1.07 | 0.38 | 0.11 |
| | [199.1–514.1] | [284.3–479.4] | [331.3–631.4] | [326.1–657.1] | [214.2–426.7] | [256.2–419.5] | [373.0–620.5] | [371.5–623.1] | | | |
| **DOMS** (score) | 0.5 ± 1.0 | 2.3 ± 1.8* | 1.9 ± 1.4* | 3.0 ± 1.7* | 0.9 ± 1.0 | 2.1 ± 1.8* | 2.0 ± 1.5* | 2.2 ± 1.4* | | <0.01 | |
| **PR** (score) | 17.4 ± 1.75 | 15.6 ± 1.74* | 15.2 ± 2.49* | 14.8 ± 2.63* | 17.8 ± 1.66 | 16.2 ± 1.90* | 14.7 ± 1.95* | 15.0 ± 2.05* | 1.02 | 0.39 | 0.10 |
| | [16.2–18.6] | [14.4–16.8] | [13.5–16.9] | [13.0–16.5] | [16.7–18.9] | [14.9–17.5] | [13.4–16.0] | [13.5–16.4] | | | |

Values are expressed as mean ± SD [95% confidence interval], except for DOMS that values are expressed as median ± interquartile distance (non-normally distributed).
CK, creatine kinase; DOMS, delayed onset muscle soreness; PR, perceived recovery.
*Significantly different from pre-exercise in both conditions (P<0.05).

$F_{(4,32)} = 21.502$, P = 0.001, $\eta_p^2 = 0.72$, $F_{(4,32)} = 37.562$, P = 0.001, $\eta_p^2 = 0.82$ and $F_{(4,32)} = 17.928$, P = 0.001, $\eta_p^2 = 0.69$, Fig 3B, 3C and 3D, respectively). However, $Qt_{w100}$ was slightly increased 48 and 72 h after the exercise in comparison to baseline (P = 0.001). The $Q_{tw10} \cdot Q_{tw100}^{-1}$ decreased significantly from baseline to immediate post-exercise (-25%) and recovered fully 24 h later (main effect of time, $F_{(4,32)} = 17.794$, P = 0.001, $\eta_p^2 = 0.68$, Fig 3E). The VA did not change throughout the experiment (main effect of time, $F_{(4,24)} = 2.297$, P = 0.088, $\eta_p^2 = 0.27$, Fig 3F). The $M_{wave}$ amplitude increased significantly from baseline to post-exercise and returned to baseline values 24 h later (main effect of time, $F_{(4,32)} = 3.081$, P = 0.029, $\eta_p^2 = 0.27$, Fig 3G). Caffeine had no effect on any neuromuscular function parameters (P > 0.05).

## Countermovement jump and performance during the sprint training

There was a significant supplement *vs.* time interaction for CMJ performance. The height ($F_{(4,28)} = 3.299$, P = 0.024, $\eta_p^2 = 0.32$, Fig 4A) and power ($F_{(4,28)} = 2.885$, P = 0.040, $\eta_p^2 = 0.29$, Fig 4B) during the CMJ were higher 24 h after exercise compared to baseline and post-exercise in both supplements (P = 0.011 and P = 0.005, respectively). However, height and power during the CMJ 48 and 72 h after exercise remained above baseline values only when caffeine was ingested (P = 0.001). The height and power during the CMJ 48 and 72 h after exercise was also higher in caffeine compared to the placebo (P = 0.010).

Pre-test sprint performance did not differ between placebo and caffeine (30 m: 6.78 ± 0.24 vs. 6.72 ± 0.22 m.s$^{-1}$, P = 0.372; 50 m: 7.48 ± 0.10 vs. 7.58 ± 0.06 m.s$^{-1}$, P = 0.632; 100 m: 7.89 ± 0.15 vs. 7.98 ± 0.10 m.s$^{-1}$, P = 0.392). In addition, sprint performance for all distances (30m, 50m and 100m) was not affected by supplement ($F_{(1,9)} = 0.083$, P = 0.779, $\eta_p^2 = 0.009$; $F_{(1,9)} = 0.203$, P = 0.663, $\eta_p^2 = 0.022$ and $F_{(1,9)} = 0.085$, P = 0.776, $\eta_p^2 = 0.009$, Table 2). However, for both conditions, sprint performance in 50 m was faster at 72 h compared to 24 h and 48 h after exercise ($F_{(2,18)} = 5.395$, P = 0.014, $\eta_p^2 = 0.374$) and faster in 100 m at 72 h compared to 48 h after exercise ($F_{(2,18)} = 6.367$, P = 0.008, $\eta_p^2 = 0.414$). There was no time effect for 30 m sprint ($F_{(2,18)} = 3.172$, P = 0.066, $\eta_p^2 = 0.260$).

## Discussion

The results of the present study indicate that although markers of muscle damage and DOMS (CK and pain feeling) remained elevated over 72 h after an exercise with eccentric emphasis, markers of central and peripheral fatigue returned to baseline values within 24 h. Caffeine did not affect neuromuscular function, but jump performance was improved with caffeine 48 and 72 h after an exercise with eccentric emphasis.

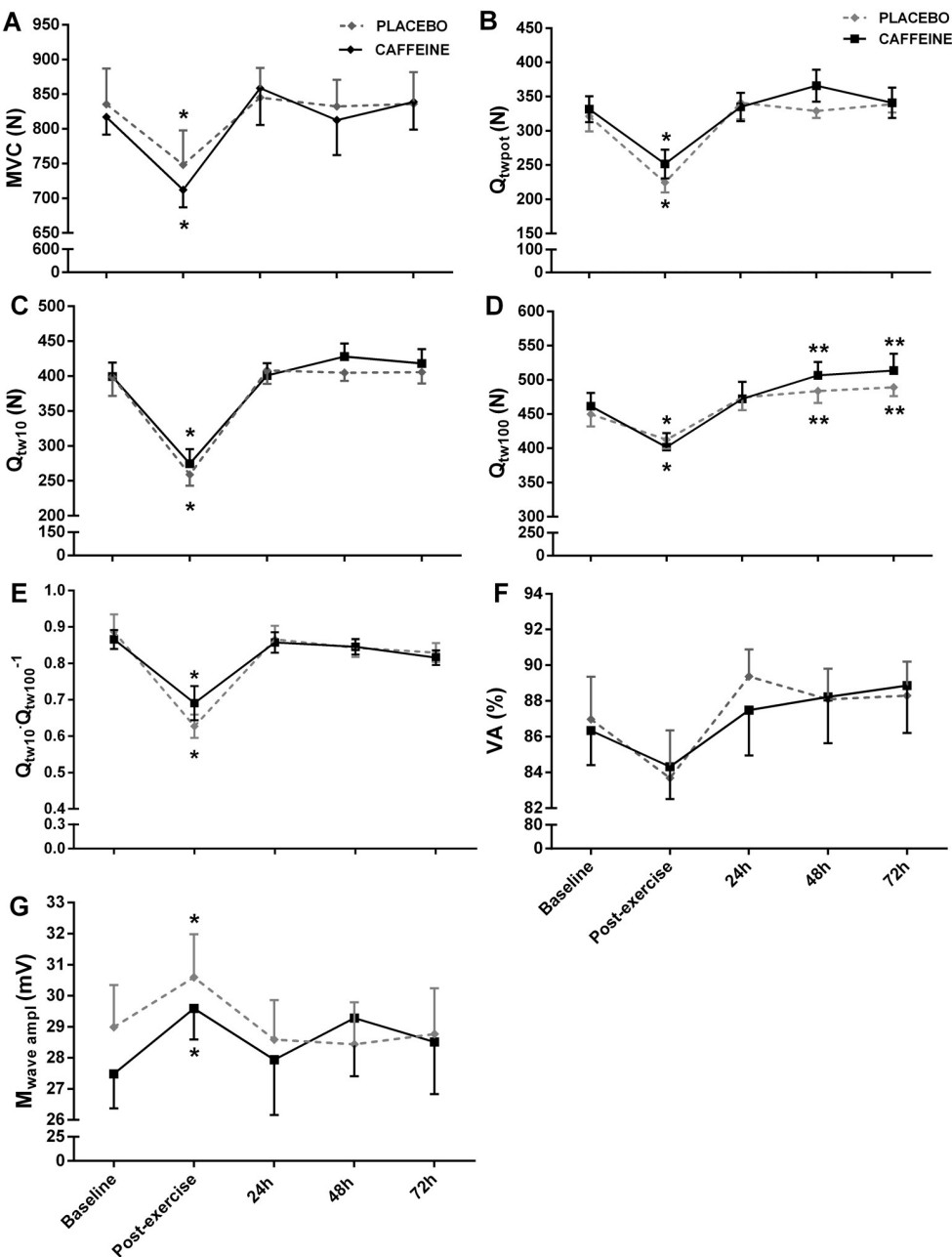

**Fig 3. Neuromuscular function assessed before and after a half-squat exercise, and at 24, 48 and 72 h after placebo or caffeine ingestion.** Values are expressed as mean ± SEM. A: MVC, maximal voluntary contraction; B: VA, voluntary activation; C: $M_{wave\ ampl}$, maximal M-wave amplitude; D: $Q_{twpot}$ single stimulation (potentiated twitch); E: $Q_{tw10}$ paired stimulations of 10 Hz; F: $Q_{tw100}$ paired stimulations of 100Hz. G: $Q_{tw10} \cdot Q_{tw100}^{-1}$ ratio. *Significantly lower than pre, 24, 48 and 72 h post half-squat exercise under both conditions; **Significantly higher than pre- and post half-squat exercise under both conditions.

The CK serum concentration increased 48 h after exercise with eccentric emphasis and remained elevated until 72 h. However, DOMS increased 24 h after exercise with eccentric emphasis and remained above baseline levels until 72 h, which coincided with a reduction in the perception of recovery. The CK after an eccentric exercise peaks 1–4 days and can remain elevated for several days [3,24,25]. An important issue is that daily training results in persistent

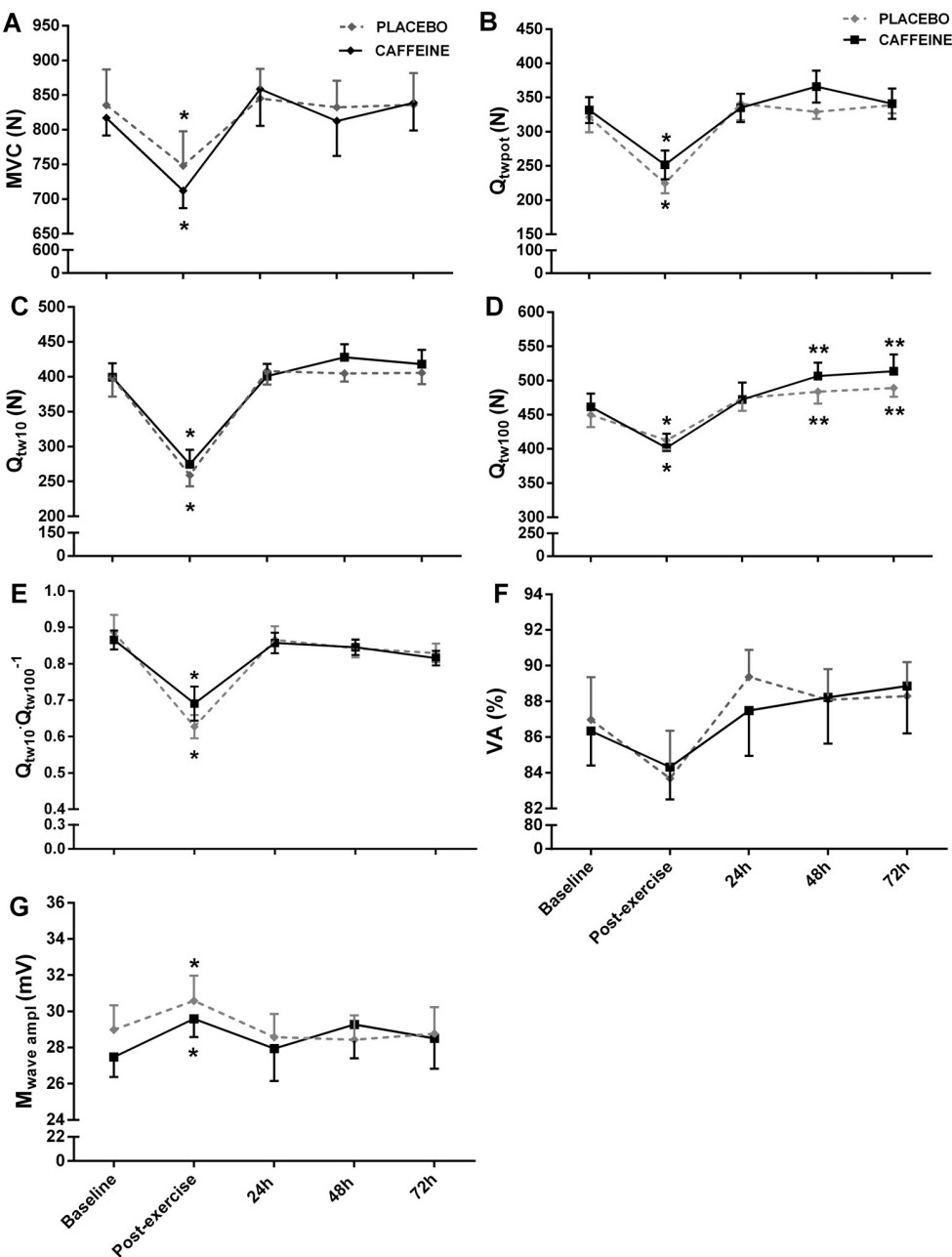

**Fig 4. Countermovement jump test performed before and after a half-squat exercise and at 24, 48 and 72 h after placebo or caffeine ingestion.** Values are expressed as mean ± SEM. A: Height; B: Power (relative to body mass). *Significantly higher than pre- and post half-squat exercise in both conditions; #Significantly higher than pre- and post half-squat exercise only in caffeine condition; †Significantly higher than placebo at the same time point.

CK elevation in athletes, with resting values being higher than in non-athletes [25]. In the present study, even providing two days of low-intensity training before taking blood sample, we found baseline CK values ranging from 199 to 514 U/L, which are slightly above the upper reference limit established to general population (174 U/L) [26]. Because it is not possible to maintain athletes without any kind of training for days, a "true" CK baseline value is hard to find in athletes. However, the values found in the present study are in agreement with

**Table 2. Sprint performance at 24, 48 and 72 h later a half-squat exercise with placebo or caffeine ingestion.**

| | PLACEBO | | | CAFFEINE | | | *F* | p | $\eta_p^2$ |
|---|---|---|---|---|---|---|---|---|---|
| | 24h | 48h | 72h | 24h | 48h | 72h | | | |
| **30m** (m.s$^{-1}$) | 6.56 ± 0.45 | 6.75 ± 0.39 | 6.65 ± 0.27 | 6.70 ± 0.42 | 6.78 ± 0.32 | 6.55 ± 0.37 | 1.29 | 0.29 | 0.12 |
| | [6.25–6.86] | [6.48–7.01] | [6.46–6.83] | [6.41–6.97] | [6.56–7.00] | [6.28–6.82] | | | |
| **50m** (m.s$^{-1}$) | 7.49 ± 0.43 | 7.54 ± 0.55 | 7.26 ± 0.33* | 7.42 ± 0.36 | 7.49 ± 0.38 | 7.27 ± 0.34* | 0.36 | 0.70 | 0.03 |
| | [7.19–7.78] | [7.16–7.91] | [7.04–7.48] | [7.17–7.67] | [7.23–7.74] | [7.02–7.51] | | | |
| **100m** (m.s$^{-1}$) | 7.89 ± 0.35 | 7.88 ± 0.38 | 7.81 ± 0.39** | 7.85 ± 0.30 | 8.05 ± 0.30 | 7.80 ± 0.33** | 1.71 | 0.20 | 0.15 |
| | [7.64–8.12] | [7.61–8.14] | [7.55–8.07] | [7.56–8.14] | [7.75–8.34] | [7.44–8.14] | | | |

Values are expressed as mean ± SD [95% confidence interval].

*Significantly different from 24 h and 48 h after exercise in both conditions (P<0.05).

**Significantly different from 48 h after exercise in both conditions (P<0.05).

reference values proposed for male athletes (82–1083 U/L) [25]. We also found an increase in CK levels 48 h after exercise, regardless of the supplement ingested. This finding is in agreement with a study showing that CK peaks within 48 h after a marathon [24]. However, our CK level 48 h after the half-squat exercise was much lower than those reported 24 h after a marathon (434–844 U/L) [24]. This is expected because CK levels after prolonged exercise such as marathon can reach up to 50 times the rest values due to the greater muscle damage caused by this kind of activity [24].

The exercise did not influence central fatigue (VA), but induced to a peripheral fatigue, as evidenced by the large reduction in $Q_{twpot}$, $Q_{tw10}$ and $Q_{tw100}$ immediately post-exercise (-30, -35 and -12%, respectively) and a low-frequency fatigue, as evidenced by a reduction in $Q_{tw10} \cdot Q_{tw100}^{-1}$ (-25%). However, peripheral and low-frequency fatigue returned to baseline levels 24 h after the exercise. The time course of neuromuscular fatigue after a eccentric-based exercise is largely variable, with studies reporting that central fatigue returned to baseline within 24 to 96 h, while peripheral fatigue within 24 to 192 h [4–8,10]. Different protocols of eccentric exercise may take in account for these inconsistencies. In the present study, we optioned for an eccentric exercise nearer to that used in the athlete's training routine. In addition, to avoid injuries, eccentric exercises used in regular training program are not designed to generate an elevated degree of muscle damage and DOMS. Our results suggest that although the considerable degree of peripheral fatigue after an exercise with eccentric emphasis (~ 30%), well-trained athletes accustomed to eccentric training can quickly restore their capacity to produce force (~24 h).

Caffeine had no influence on CK, DOMS or neuromuscular function. However, the height and power during the CMJ returned to baseline levels 48 h after the exercise with the placebo, while the height and power was maintained above baseline levels until 72 h after the exercise with caffeine. Muscle power was also higher in caffeine than in placebo at 48 and 72 h post-exercise. Previous studies have shown improved jump performance after the ingestion of 3 to 6 mg.kg$^{-1}$ of caffeine [27–32]. Recent reviews concluded that caffeine promotes an ergogenic effect on muscle strength and power [23,33,34]. However, no study to date has demonstrated this improvement even after an exercise with eccentric emphasis using high performance sprinters and jumpers. Nevertheless, the improved power was not translated to an improved sprint performance in the present study. Similar findings have been reported showing no improvement in repeated-sprint ability with caffeine ingestion [30]. The shorter duration of the contraction and the simplicity of the technique may explain why performance was improved in the CMJ but not in the sprint with caffeine [30].

Our study demonstrated that a half-squat exercise session with an eccentric emphasis induces peripheral but not central fatigue, which is restored within 24 hours in well-trained jumpers and sprinters. Anhydrous caffeine (5 mg.kg$^{-1}$) improved jump performance 48 and 72 h after a half-squat exercise with an emphasis on eccentric action. However, neuromuscular function and sprint performance were not influenced by caffeine intake. Our results has considerable practical relevance as they are indicating that caffeine can optimize jumping performance in well-trained athletes even when a certain degree of muscle damage and DOMS are present. Thus, ingestion of anhydrous caffeine in some jump training sessions may be an interesting strategy to improve training quality and performance of these athletes.

There are some limitations in the present study that should be mentioned. Our resistance training protocol generated mild to moderate muscle pain, resulting in no impairment of subsequent training capacity. Thus, whether caffeine would be useful when greater muscle pain is present deserves further investigation. Another potential limitation is that experimental conditions were performed only once each (one for placebo and another one to caffeine). Repetitions of the experimental blocks may have provided additional information regarding the reproducibility of our findings.

## Conclusions

In conclusion, our study demonstrates that even with a certain low degree of muscle damage and DOMS over 72 h after an exercise with eccentric emphasis, neuromuscular function, muscle strength and sprint performance are preserved in well-trained sprinters and jumpers. Caffeine ingestion (5 mg.kg$^{-1}$) improves muscle power 48 and 72 h after an exercise with eccentric emphasis, but it has no effect on neuromuscular function and sprint performance.

## Acknowledgments

The authors are thankful for all athletes who took part of the study and for the technical assistance of the staffs of the Regional Center for Initiation of Athletics from Lavras. The authors thank Dr. Marcos Silva-Cavalcante for his critical suggestions and contribution to the paper.

## Author Contributions

**Conceptualization:** Ana C. Santos-Mariano, Fernando R. De-Oliveira, Adriano E. Lima-Silva.

**Data curation:** Ana C. Santos-Mariano, Fabiano Tomazini.

**Formal analysis:** Ana C. Santos-Mariano, Fabiano Tomazini, Leandro C. Felippe, Daniel Boari, Adriano E. Lima-Silva.

**Investigation:** Ana C. Santos-Mariano, Fabiano Tomazini.

**Methodology:** Ana C. Santos-Mariano, Fernando R. De-Oliveira, Adriano E. Lima-Silva.

**Project administration:** Adriano E. Lima-Silva.

**Supervision:** Fernando R. De-Oliveira, Adriano E. Lima-Silva.

**Writing – original draft:** Ana C. Santos-Mariano, Adriano E. Lima-Silva.

**Writing – review & editing:** Ana C. Santos-Mariano, Romulo Bertuzzi, Adriano E. Lima-Silva.

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
