## [Decision Letter · Decision Letter 0]

20 Aug 2019

PONE-D-19-16976

Effect of caffeine on neuromuscular function following eccentric-based exercise

PLOS ONE

Dear Dr. Lima-Silva,

Thank you for submitting your manuscript to PLOS ONE. After careful consideration, we feel that it has merit but does not fully meet PLOS ONE’s publication criteria as it currently stands. Therefore, we invite you to submit a revised version of the manuscript that addresses the points raised during the review process.

PLoS ONE policies do not address originality or novelty of a manuscript for being considered for publication. However, it is mandatory that the work was performed with appropriate methodology and without major flaws. Therefore, I would like to ask you have special care when adjusting the manuscript following both reviewers' concerns.

We would appreciate receiving your revised manuscript by Sep 29 2019 11:59PM. To enhance the reproducibility of your results, we recommend that if applicable you deposit your laboratory protocols in protocols.io, where a protocol can be assigned its own identifier (DOI) such that it can be cited independently in the future. For instructions see: http://journals.plos.org/plosone/s/submission-guidelines#loc-laboratory-protocols

We look forward to receiving your revised manuscript.

Kind regards,

Daniel Boullosa

Academic Editor

PLOS ONE

Journal Requirements:

Fabiano Tomazini is grateful for his scholarship received from the Foundation to Support Science and Research in Pernambuco State (FACEPE-Brazil, process number: IBPG-1716-4.05/15). This study was financed in part by the Coordenação de Aperfeiçoamento de Pessoal de Nível Superior - Brazil (CAPES) - Finance Code 001.

This study was financed in part by the Coordenação de Aperfeiçoamento de Pessoal de Nível Superior - Brazil (CAPES) - Finance Code 001

Reviewers' comments:

Reviewer's Responses to Questions

**Comments to the Author**

1. Is the manuscript technically sound, and do the data support the conclusions?

Reviewer #1: Yes

Reviewer #2: Partly

2. Has the statistical analysis been performed appropriately and rigorously? 

Reviewer #1: Yes

Reviewer #2: Yes

3. Have the authors made all data underlying the findings in their manuscript fully available?

Reviewer #1: Yes

Reviewer #2: No

4. Is the manuscript presented in an intelligible fashion and written in standard English?

Reviewer #1: Yes

Reviewer #2: Yes

5. Review Comments to the Author

Reviewer #1: Manuscript Number: PONE-D-19-16976

Effect of Caffeine on neuromuscular function following eccentric-based exercise

General Comments

Authors carried out an interesting research regarding to the effect of caffeine on recovery of neuromuscular function, power and sprint performance following an eccentric-based exercises using a "Tirante Muscular� " in half squat exercise. Authors did a great job controlling many variables providing a broad vision when studying central and peripheral fatigue, DOMS, perceived recovery, muscle damage, CMJ and Sprint. However, the manuscript is well written and very good organized. Should be highlight why the Tirante Muscular� has been used, since it is not clear the reason that you chose this tool and not a Smith Machine or free weight. On the other hand, in the paragraph of experimental design, authors describe the sprint pre-test as “one sprint of 30 m, one of 50 m and one of 100m”. In the post- test, has not been carried out the same procedure, conversely have been performed: “1) 3 sets of 30m sprints, with a 5min rest between sets; 2) 3 sets of 50 m sprints, with a 7 min rest between sets and; 3) 3 sets of 100 m sprints, with a 10 min rest between sets”. These two protocols are different and the results cannot be compared due to the volume and the fatigue in each test. It is a limitation of this study that should be shown in the manuscript. It is possible to think that the performance in the sprint has not improved due to the fatigue that the post-test supposes. Perhaps a single series of 30m would have been enough to compare the performance in the sprint effectively. By this is important that authors explain it. Finally, only some changes I can propose to improve this manuscript. Thus, I considered that the paper could be accepted if the authors could correct mistakes pointed out. My specific comments are presented below.

Specific Comments

Page 2, line 42: Before “and”, “,” should not be used. On the other hand, between “72” and “h” the “-” it is not necessary and the authors due write in the same form along the manuscript because in some case don’t use hyphen.

Page 2, line 45: The use of concept baseline is not appropriated to refer to Sprint performance. This concept is well used to muscle damage, DOMS or RT, but sprint need a warm up and activation process to do the best performance. By this, change “Baseline” by “Pre-test” could be more appropriated.

Page 2, line 48,52: Remove the hyphen (24 h) (48 h) (72 h).

Page 3, line 73: It would be good to write a brief description of how DOMS is measured in the literature.

Page 3, line 73,77,78: Remove the hyphen (24 h) (48 h) (72 h).

Page 4, line 101: It is necessary write the name of Creatine Kinase before (CK) since is the first time that appear this concept during de manuscript.

Page 5, line 111-112: After “years old” should be used colon not comma and it is necessary write “height:” before 180.7…. Also, it is interesting to report the Mean and SD of 1RM load or the load used corresponding with 70%1RM.

Page 5, line 123: The sentence starts very similar like the sentence before. Change “study” by “research” could be a good strategy of avoid be repetitive.

Page 5, line 126: Explain if the 1RM test was carried out with Tirante Muscular� .

Page 6, line 133-134: Unify the way to refer days (with numbers “2” or letters “two”).

Page 6, line 134: Change “baseline” by “Pre”. Also, before “and”, “,” should not be used, remove it.

Page 6, line 134-135: How much time was used to recover between the sprints?

Page 6, line 134-137: Was carried out some warm up before Sprints and CMJ? Could you explain it?

Page 6, line 139: Explain if the half-squat exercise was performed with Tirante Muscular�.

Page 6, line 157: Remove the hyphen (48 h).

Page7, line 173,175: Remove the hyphen (2 min).

Page 8, line 182,183: Remove the hyphens (2 min) (3 s).

Page 8, line 193: Remove the hyphen (0.5 min).

Page 8, line 202: Remove the hyphen (30 s).

Page 9, line 209,211,212,216: Remove the hyphens.

Page 10, line 240: Could you explain the stretching exercises that was carried out? It is known that passive stretching can impair the performance.

Page 11, line 251: Explain if a nurse or another qualified professional took the serum extractions.

Page 11, line 252: You have used “Creatine Kinase” before, the abbreviation is enough in this case.

Page 11, line 258, 260: Unify the way to refer the numbers of the scale (with numbers “0” or letters “zero” and “six”). Change words by numbers.

Page 12, line 284,285: Before “and”, “,” should not be used, remove it.

Page 12, line 289: If you have used “a half-squat exercise” during all manuscript in this case is more appropriate than “a half-squat resistance training”. Change it.

Page 13, line 299: Before “and”, “,” should not be used, remove it.

Page 13, line 312: Add “exercise” after a half squat.

Page 14, line 333: You can use “CMJ” instead of “countermovement jump”

Page 14, line 334: If you have used “a half-squat exercise” during all manuscript in this case is more appropriate than “a half-squat resistance training”. Change it.

Page 15, line 346: If you have used “a half-squat exercise” during all manuscript in this case is more appropriate than “a half-squat resistance training”. Change it.

Page 16, line 368: Add “s” after “athletes’ ”

Page 16, line 378: Remove the hyphens (48 h) and (72 h)

Page 17, line 392: It would be interesting to complete the paragraph with a section of limitations, perhaps report about the pre and post sprints test or regarding to only use one caffeine intake protocol.

Figures 3 and 4: Unify the levels of values in the Y axis of all figures. MVC has nine levels, Qtwpost has seven levels or Qtw100 has eight levels.

Reviewer #2: Review to: PONE-D-19-16976; Effect of caffeine on neuromuscular function following eccentric-based exercise.

Major comments.

This experiment was designed to assess the neuromuscular effects of caffeine intake during the recovery phase of an eccentric-based exercise protocol that induces muscle damage. Overall, the experiment seems well designed, it contains several measurements to fulfill the objectives of the investigation and it is conveniently described in a well-written manuscript. However, the experiment has a serious flaw: it is designed to assess caffeine’s effects in the recovery phase of a muscle-damaging exercise when most, if not all, performance variables returned to basal -non-fatigued- values within 24 hours after the end of exercise. This fact is likely due to the use of a “light” eccentric exercise protocol that produced very low levels of muscle damage – a pilot study would have been helpful to detect the inefficacy of this protocol to produce moderate muscle damage”.

Thus, although I found merit in the experiment and I congratulate authors for the hard work developed for this investigation, I would suggest that authors reconsider the use of “recovery” through the manuscript because this is not what they were investigating -the recovery lasted < 24 h and they did not perform any measurement in this time. For example, the conclusion of the abstract should indicate that:

“Caffeine improves muscle power 48 and 72 hours after an eccentric-based exercise…”.

A limitation paragraph including this aspect is highly recommended.

Other comments.

Introduce information about the use of caffeine in athletics. Is this common?

Indicate why you selected this protocol of eccentric exercise. And discuss the lack of a relevant increase in CK (for instance, you could compare these values with the ones found after a marathon).

Indicate the habituation to caffeine of these individuals.

The CK concentrations before exercise seems quite high (especially one participant with > 500 U/L). Please, discuss this in the light of previous investigations. Did they have previous muscle damage due to training?

Why there is no data on jump height?

Please, indicate the calculation of the sample size or at least, the statistical power with this sample of 11 individuals.

A paragraph of practical applications for jumpers and sprinter might be suitable. How they should use caffeine during their training. Before all sessions? 48 h after a high-intensity session?

Line 397. “sprinters” is twice

6. PLOS authors have the option to publish the peer review history of their article (what does this mean?). If published, this will include your full peer review and any attached files.

Reviewer #1: Yes: González-Hernández Jorge Miguel

Reviewer #2: No

---

## [Author Response · Author response to Decision Letter 0]

19 Sep 2019

September 16, 2019

Dear Dr. Daniel Boullosa

Academic Editor of the PLOS ONE

We thank the referees for their time and effort spent in carefully reviewing our manuscript and providing such valuable comments. We also thank the editor for the opportunity to revise our manuscript: PONE-D-19-16976,

“Effect of caffeine on neuromuscular function following eccentric-based exercise”. We carefully considered all referees’ suggestions and included corresponding alterations in the manuscript. Revised points in the manuscript are highlighted in yellow and a point-by-point reply is detailed below. We believe all these suggestions have improved the quality of our manuscript.

Reviewer #1

Authors did a great job controlling many variables providing a broad vision when studying central and peripheral fatigue, DOMS, perceived recovery, muscle damage, CMJ and Sprint. However, the manuscript is well written and very good organized. Should be highlight why the Tirante Muscular� has been used, since it is not clear the reason that you chose this tool and not a Smith Machine or free weight. On the other hand, in the paragraph of experimental design, authors describe the sprint pre-test as “one sprint of 30 m, one of 50 m and one of 100m”. In the post- test, has not been carried out the same procedure, conversely have been performed: “1) 3 sets of 30m sprints, with a 5min rest between sets; 2) 3 sets of 50 m sprints, with a 7 min rest between sets and; 3) 3 sets of 100 m sprints, with a 10 min rest between sets”. These two protocols are different and the results cannot be compared due to the volume and the fatigue in each test. It is a limitation of this study that should be shown in the manuscript. It is possible to think that the performance in the sprint has not improved due to the fatigue that the post-test supposes. Perhaps a single series of 30m would have been enough to compare the performance in the sprint effectively. By this is important that authors explain it.

Response: Thank you for these positive comments. We appreciate your suggestions to improve our paper. Regarding the Tirante Muscular®, we have chosen this device to increase the eccentric action during the half-squat exercise. This device is largely used by athletes in their training routine because they do not need to use excessive high external loads to increase eccentric action, avoiding injuries during eccentric-based training. A previous study showed that eccentric action is considerably increased by using this device (EDIR DA SILVA et al., 2005; http://www.redalyc.org/articulo.oa?id=551656963007). We have added this information in the manuscript on Page 8, Line 179 as follow:

“This flexible strip has been demonstrated suitable to increase eccentric action during a half-squat exercise (17). Participants were familiar with this accessory as they had used it in their training routine as a form to increase eccentric action without using an excessive external load, which may reduce the risk of injuries associated with eccentric-based training, mainly in a preparatory phase of an annual training periodization (17)”. 

Regarding the sprint training performance, we agree that pre-test cannot be directly compared to 24h, 48h and 72h recovery. We performed the pre-test sprints to check whether athletes would be fully recovered at the transition from the first experimental block (using placebo or caffeine) to the second experimental block (when supplement was inverted). Only one repetition of each distance was performed to avoid the development of fatigue and to not influence the following days tests. This procedure guaranteed that all blocks (placebo and caffeine) started with participants having fully recovered from previous tests and training and with the same recovery level in both conditions. For the post-test (i.e., sprints at 24, 48 and 72h), more series were performed to simulate a typical sprint training session and to understand whether caffeine would be helpful to improve sprint training performance. We agree these assumptions were not clear in the paper and our statistical analysis might have not fit well to this purpose. Thus, in this new version we have compared the two pretests sprint performance just to check if there was difference in the recovery from an experimental block to another, and then separately compared the sprint performance between placebo and caffeine over the posttest period. We have included this information on Page 13, Line 294: “The pre-test sprint performance before the first and second experimental blocks were compared using a paired t test in order to check if athletes had fully recovered from the previous tests and training sessions. The performance during the sprint training was then analyzed separately using a two-way, repeated-measures ANOVA [supplement (PLA and CAF) x time (24, 48 and 72 h post exercise)”. 

 Page 16, Line 362 “Pre-test sprint performance did not differ between placebo and caffeine (30 m: 6.78 ± 0.24 vs. 6.72 ± 0.22 m.s-1, P = 0.372; 50 m: 7.48 ± 0.10 vs. 7.58 ± 0.06 m.s-1, P = 0.632; 100 m: 7.89 ± 0.15 vs. 7.98 ± 0.10 m.s-1, P = 0.392). In addition, sprint performance for all distances (30m, 50m and 100m) was not affected by supplement (F(1,9) = 0.083, P = 0.779 , ηp2 = 0.009; F(1,9) = 0.203 , P = 0.663, ηp2 = 0.022 and F(1,9) = 0.085, P = 0.776 , ηp2 = 0.009, Table 2). However, for both conditions, sprint performance in 50 m was faster at 72 h compared to 24 h and 48 h after exercise (F(2,18) = 5.395 , P = 0.014, ηp2 = 0.374) and faster in 100 m at 72 h compared to 48 h after exercise (F(2,18) = 6.367, P = 0.008 , ηp2 = 0.414). There was no time effect for 30 m sprint (F(2,18) = 3.172, P = 0.066 , ηp2 = 0.260)”.

Minor: 

Page 2, line 42: Before “and”, “,” should not be used. On the other hand, between “72” and “h” the “-” it is not necessary and the authors due write in the same form along the manuscript because in some case don’t use hyphen.

Response: Thank you for this recommendation. We have modified accordingly throughout the manuscript. 

Page 2, line 45: The use of concept baseline is not appropriated to refer to Sprint performance. This concept is well used to muscle damage, DOMS or RT, but sprint need a warm up and activation process to do the best performance. By this, change “Baseline” by “Pre-test” could be more appropriated.

Response: We have changed “baseline” for “pre-test” when referring to sprint throughout the text, as suggested. 

Page 2, line 48,52: Remove the hyphen (24 h) (48 h) (72 h).

Response: We have removed accordingly (Page 2, Line 48, 52). 

Page 3, line 73: It would be good to write a brief description of how DOMS is measured in the literature.

Response: This has been added on Page 12, Line 273 “The DOMS was measured with the Pain Intensity Scale ranging from 0 (no pain) to 10 (very intense pain, almost unbearable) (9)”. 

Page 3, line 73,77,78: Remove the hyphen (24 h) (48 h) (72 h).

Response: We have removed accordingly (Page 3, Line 74, 77, 79, 80).

Page 4, line 101: It is necessary write the name of Creatine Kinase before (CK) since is the first time that appear this concept during de manuscript. 

Response: We thank the reviewer to pick up this mistake. The CK has been defined now in his first apparition “… creatine kinase (CK), a marker of muscle damage”. (Page 3, line 66).

Page 5, line 111-112: After “years old” should be used colon not comma and it is necessary write “height:” before 180.7…. Also, it is interesting to report the Mean and SD of 1RM load or the load used corresponding with 70%1RM.

Response: We thank the reviewer to pick up these typo error. This part of the text is now reading “…(age: 18.7 ± 2.7 years old; weight: 69.9 ± 6.4 kg; height: 180.7 ± 7.7 cm)”. (Page 5, line 115-116). 

The 1RM load and the load corresponding to 70% of 1 RM were also inserted:

“The 1RM load was 130.2 ± 18.6 kg”. (Page 8, line 188).

“The load at 70% of 1RM was 91.1 ± 13.9 kg”. (Page 9, line 204).

Page 5, line 123: The sentence starts very similar like the sentence before. Change “study” by “research” could be a good strategy of avoid be repetitive. 

Response: The first sentence has been changed to “(The research…)” (Page 5, line 123).

Page 5, line 126: Explain if the 1RM test was carried out with Tirante Muscular� .

Response: We have included this information accordingly “Two preliminary sessions were designed to assess one-repetition maximal strength in a half-squat exercise (1RM) using a Tirante Musculador® (see one-repetition maximal strength test section for details)”. (Page 5, line 130-132).

Page 6, line 133-134: Unify the way to refer days (with numbers “2” or letters “two”).

Response: This sentence has been changed to “After two days of low-intensity training, premeasurements were performed over the next two days”. (Page 6, line 138-139).

Page 6, line 134: Change “baseline” by “Pre”. Also, before “and”, “,” should not be used, remove it.

Response: We have modified accordingly. This sentence has been changed to “premeasurements were performed…”. (Page 6, line 138).

Page 6, line 134-135: How much time was used to recover between the sprints?

Response: We have added this information “pre-test sprints (one sprint of 30 m following by 5 min rest, one sprint of 50 m following by 7 min rest and one sprint of 100 m)”. (Page 6, line 140-141).

Page 6, line 134-137: Was carried out some warm up before Sprints and CMJ? Could you explain it?

Response: Athletes performed a 20 min warm-up (10 minutes running + 10 minutes of dynamic stretching) before sprints. This information has been included on Page 6, Line 139 “On day 1, after a 20 min warm-up (10 minutes running + 10 minutes of dynamic stretching)”. Because athletes performed the CMJ immediately after the neuromuscular function assessment, and neuromuscular assessment was precluded by a warm-up, athletes were already warmed up and no additional warm-up was required before CMJ. Details of the warm-up before neuromuscular function assessment is provided on Page 6, Line 144 “Neuromuscular function was preceded by a warm-up (see neuromuscular function assessment section for details)”. 

Page 9, Line 224-227 “The neuromuscular function assessment (except immediately after the half-squat exercise) was preceded by a warm up composed of a 5 min jogging and 4 x 5 s isometric contractions of knee extension at ~60% of maximal subjective isometric force (30 s rest between contractions)”.

Page 6, line 139: Explain if the half-squat exercise was performed with Tirante Muscular�.

Response: We have included accordingly “a half-squat exercise with emphasis on eccentric action were performed using the Tirante Musculador®”. (Page 6, line 146-147).

Page 6, line 157: Remove the hyphen (48 h).

Response: This has been removed accordingly (Page 7, line 167).

Page7, line 173,175: Remove the hyphen (2 min).

Response: This has been removed accordingly (Page 8, line 185, 187).

Page 8, line 182,183: Remove the hyphens (2 min) (3 s).

Response: This has been removed accordingly (Page 8, line 196,197).

Page 8, line 193: Remove the hyphen (0.5 min).

Response: This has been removed accordingly (Page 9, line 211).

Page 8, line 202: Remove the hyphen (30 s).

Response: This has been removed accordingly (Page 9, line 219).

Page 9, line 209,211,212,216: Remove the hyphens.

Response: This has been removed accordingly (Page 9-10, line 225, 226, 228, 231).

Page 10, line 240: Could you explain the stretching exercises that was carried out? It is known that passive stretching can impair the performance. 

Response: We were aware that passive stretching impairs sprint performance; therefore, athletes performed dynamic stretching exercises, as they usually do before their training and competition. Dynamic stretching is recommended as warm up for sprint due to the similarity of the movement patterns, the consequent increase in body temperature and proprioception aid (WINCHESTER et al., 2008; doi: 10.1519/JSC.0b013e31815ef202). We have added that dynamic stretching was used as warm up “The sprint training session started with a 20 min warm-up (10 minutes running + 10 minutes of dynamic stretching)”. (Page 11, line 253).

Page 11, line 251: Explain if a nurse or another qualified professional took the serum extractions. 

Response: All blood samples were collected by an accredited professional phlebotomist. We have included this information “Blood samples (8 ml) were drawn by a professional phlebotomist from the antecubital vein by venipuncture…”. (Page 11, line 264).

Page 11, line 252: You have used “Creatine Kinase” before, the abbreviation is enough in this case. 

Response: We thank the reviewer to pick up this mistake. The sentence now reads “The serum CK concentration was determined…”. (Page 11, line 266).

Page 11, line 258, 260: Unify the way to refer the numbers of the scale (with numbers “0” or letters “zero” and “six”). Change words by numbers.

Response: This has been changed accordingly “The DOMS was measured with the Pain Intensity Scale ranging from 0 (no pain) to 10 (very intense pain, almost unbearable) (9). The perception of recovery was measured on the Total Quality of Recovery Scale ranging from 6 to 20, where 6 is "nothing recovered" and 20 is "fully recovered”. (Page 12, line 273-276).

Page 12, line 284,285: Before “and”, “,” should not be used, remove it.

Response: We have removed accordingly (Page 12, line 290,291).

Page 12, line 289: If you have used “a half-squat exercise” during all manuscript in this case is more appropriate than “a half-squat resistance training”. Change it. 

Response: The sentence now reads “Creatine kinase, delayed onset muscle soreness and perceived recovery at baseline, and at 24, 48 and 72 h later a half-squat exercise” . (Page 14, line 316).

Page 13, line 299: Before “and”, “,” should not be used, remove it.

Response: We have removed accordingly (Page 14, line 329).

Page 13, line 312: Add “exercise” after a half squat.

Response: We have added accordingly. The text now reads “Neuromuscular function assessed before and after a half-squat exercise”. (Page 15, line 341).

Page 14, line 333: You can use “CMJ” instead of “countermovement jump”

Response: The sentence now reads “Countermovement jump test performed before and after a half-squat exercise”. (Page 16, line 372).

Page 14, line 334: If you have used “a half-squat exercise” during all manuscript in this case is more appropriate than “a half-squat resistance training”. Change it. 

Response: We have changed accordingly. This text now reads “. Countermovement jump test performed before and after a half-squat exercise”. (Page 16, line 372).

Page 15, line 346: If you have used “a half-squat exercise” during all manuscript in this case is more appropriate than “a half-squat resistance training”. Change it. 

Response: We have changed accordingly. This text now reads. “Sprint performance at 24, 48 and 72 h later a half-squat exercise”. (Page 17, line 382).

Page 16, line 368: Add “s” after “athletes’ ”

Response: We have modified accordingly (Page 18, line 422).

Page 16, line 378: Remove the hyphens (48 h) and (72 h)

Response: This has been removed (Page 19, line 432).

Page 17, line 392: It would be interesting to complete the paragraph with a section of limitations, perhaps report about the pre and post sprints test or regarding to only use one caffeine intake protocol. 

Response: A new paragraph has been added to address limitations. We have added the low level of pain generated by our resistance training protocol (suggestion of reviewer 2) and using only one caffeine intake protocol (your suggestion) as limitations. However, based in your first comment, we have reanalyzed the sprint data and felt we have solved the problem of the differences in the numbers of sprints between pre and post sprints (please, see our answer for your first comment). Thus, we think this is now not properly a limitation. The limitation paragraph reads (Page 20, line 452-458) “There are some limitations in the present study that should be mentioned. Our resistance training protocol generated mild to moderate muscle pain, resulting in no impairment of subsequent training capacity. Thus, whether caffeine would be useful when greater muscle pain is present deserves further investigation. Another potential limitation is that experimental conditions were performed only once each (one for placebo and another one to caffeine). Repetitions of the experimental blocks may have provided additional information regarding the reproducibility of our findings”.

Figures 3 and 4: Unify the levels of values in the Y axis of all figures. MVC has nine levels, Qtwpost has seven levels or Qtw100 has eight levels.

Response: We have modified Y- axis in figures 3 and 4 and now all Y-axis have nine levels.

Reviewer #2:

This experiment was designed to assess the neuromuscular effects of caffeine intake during the recovery phase of an eccentric-based exercise protocol that induces muscle damage. Overall, the experiment seems well designed, it contains several measurements to fulfill the objectives of the investigation and it is conveniently described in a well-written manuscript. However, the experiment has a serious flaw: it is designed to assess caffeine’s effects in the recovery phase of a muscle-damaging exercise when most, if not all, performance variables returned to basal -non-fatigued- values within 24 hours after the end of exercise. This fact is likely due to the use of a “light” eccentric exercise protocol that produced very low levels of muscle damage – a pilot study would have been helpful to detect the inefficacy of this protocol to produce moderate muscle damage”. Thus, although I found merit in the experiment and I congratulate authors for the hard work developed for this investigation, I would suggest that authors reconsider the use of “recovery” through the manuscript because this is not what they were investigating -the recovery lasted < 24 h and they did not perform any measurement in this time. For example, the conclusion of the abstract should indicate that: “Caffeine improves muscle power 48 and 72 hours after an eccentric-based exercise…”. A limitation paragraph including this aspect is highly recommended.

Response: We are happy that you have appreciated our experiment design and the quality of writing of the paper. We also thank you for your suggestions to improve our paper. We agree that our eccentric exercise protocol was not intense enough to provoke long impairment in our main outcomes. A pilot study would be helpful, but as we recruited athletes involved in national and international competitions and they were engaged in experiments taking 3 weeks, we were afraid to propose a longer experimental design and impair their preparation and carry to decline participation. In addition, we were afraid to propose a more intense protocol to produce larger muscle damage because they were in the preparatory phase of an annual training periodization; therefore, more susceptive to muscle injuries if using intense muscle damage protocol. To not miss the opportunity to test performance in athletes of this level, we ascertain this light protocol with their coach. However, we recognize this is a limitation and the term “recovery” should not be used. Based on your suggestion, we have removed the use of recovery throughout the manuscript and replaced to “after a half-squat exercise with eccentric emphasis” or something similar. Conclusion in the abstract was also altered as suggested (Page 2, Line 54) to “In conclusion, caffeine improves muscle power 48 and 72 h after an eccentric-based exercise, but it has no effect on neuromuscular function and sprint performance”. We have also added a paragraph in the discussion recognized this point as a limitation, as suggested (Page 20, line 452-458) “There are some limitations in the present study that should be mentioned. Our resistance training protocol generated mild to moderate muscle pain, resulting in no impairment of subsequent training capacity. Thus, whether caffeine would be useful when greater muscle pain is present deserves further investigation. Another potential limitation is that experimental conditions were performed only once each (one for placebo and another one to caffeine). Repetitions of the experimental blocks may have provided additional information regarding the reproducibility of our findings”.

Based on the data presented, I have the following comments/questions.

1. Introduce information about the use of caffeine in athletics. Is this common?

Response: Thank you for this question. Yes, it was. Since caffeine has been removed from the banned list of substances of World Anti-Doping Agency (WADA), there are a progressive increase in caffeine use as an ergogenic resource by athletes of various sports, including athletics (CHESTER & WOJEK, 2008; DOI 10.1055/s-2007-989231). We have included a sentence in the introduction providing this information (Page 4, line 90-93) “Following its removal from the World Anti-Doping Agency (WADA) Prohibited List in 2004, the use of caffeine for track and field athletes increased substantially (12). The use of caffeine has been reported to be large in national/international level athletes (~48%) and in power athletes (~57%) (12)”. 

2. Indicate why you selected this protocol of eccentric exercise. And discuss the lack of a relevant increase in CK (for instance, you could compare these values with the ones found after a marathon).

Response: We have included an explanation about this choice on Page 8, Line 200-204 “We ascertain this protocol with athetes’s coach to simulate their training routine. We chose a protocol able to provoke moderate rather than a more intense DOMS because athletes were in a preparatory phase of an annual training periodization; therefore, protocol to induced more intense DOMS would make athletes more susceptive to muscle injuries”.

 We have also included in the discussion section the lack of a relevant increase in CK and the comparison with values after a marathon, as suggested (Page 18, Line 406- 412) “We also found an increase in CK levels 48 h after exercise, regardless of the supplement ingested. This finding is in agreement with a study showing that CK peaks within 48 h after a marathon (24). However, our CK level 48 h after the half-squat exercise was much lower than those reported 24 h after a marathon (434-844 U/L) (24). This is expected because CK levels after prolonged exercise such as marathon can reach up to 50 times the rest values due to the greater muscle damage caused by this kind of activity (24)”.

3. Indicate the habituation to caffeine of these individuals.

Response: We have included the athletes’ daily caffeine consumption on Page 5, Line 117 “low-to-moderate habitual caffeine consumption (< 80 mg.day-1) (16)”. 

4. The CK concentrations before exercise seems quite high (especially one participant with > 500 U/L). Please, discuss this in the light of previous investigations. Did they have previous muscle damage due to training?

Response: Thank you for this question. The baseline CK upper limit for general population is 174 U/L. We found an average baseline value of 330 U/L, which suggests a slightly already elevated CK. Unfortunately, daily training results in persistent CK elevation in athletes, which makes hard to find a “true” baseline. We tried to control this by providing two days of low-intensity training before blood sample, as it would be difficult to maintain high performance athletes without any kind of training. Although we have not got values below 174 U/L, the values were quite close (~ 330 U/L) and not different between supplements, which suggests that any effect of previous training on CK as similar across experimental conditions. We have added a discussion about this on Page 17, Line 397-406 “The CK after an eccentric exercise peaks 1–4 days and can remain elevated for several days (3,24,25). An important issue is that daily training results in persistent CK elevation in athletes, with resting values being higher than in non-athletes (25). In the present study, even providing two days of low-intensity training before taking blood sample, we found baseline CK values ranging from 199 to 514 U/L, which are slightly above the upper reference limit established to general population (174 U/L) (26). Because it is not possible to maintain athletes without any kind of training for days, a “true” CK baseline value is hard to find in athletes. However, the values found in the present study are in agreement with reference values proposed for male athletes (82-1083 U/L) (25)”.

5. Why there is no data on jump height?

Response: We have included data from jump height in this new version, as suggested. We have changed information in Methods section (Page 10, Line 246) “The CMJ height was calculated from flying time, while CMJ power calculated from flying time and body mass, using a commercial software (Jump System 1.0, Cefise, São Paulo, Brazil).” and Results section (Page 15, Line 353-360) “There was a significant supplement vs. time interaction for CMJ performance. The height (F(4,28) = 3.299, P = 0.024, ηp2 = 0.32, Fig 4A) and power (F(4,28) = 2.903, P = 0.039, ηp2 = 0.29, Fig 4B) during the CMJ were higher 24 h after exercise compared to baseline and post-exercise in both supplements (P = 0.011 and P = 0.005, respectively). However, height and power during the CMJ 48 and 72 h after exercise remained above baseline values only when caffeine was ingested (P = 0.001). The height and power during the CMJ 48 and 72 h after exercise was also higher in caffeine compared to the placebo (P = 0.010)”. We have also included CMJ height in figure 4.

6. Please, indicate the calculation of the sample size or at least, the statistical power with this sample of 11 individuals.

Response: We have added a paragraph in the statistical analysis section explaining the calculation of the sample size “Because of the lack of data regarding the caffeine effects on neuromuscular function post an eccentric-based exercise in athletes, the required sample size was estimated using an effect size (ES = 0.37) reported in a meta-analysis investigating the effect of caffeine ingestion on muscular strength (23). With an alpha of 0.05 and a desired power of 0.80, the total sample size necessary to achieve statistical significance was estimated to be 10 participants. However, the starting sample size was increased to 11 participants, assuming that 10% might drop out during the data collection. The sample size calculation was performed using G*Power software (version 3.1.9.2, Kiel University, Germany). (Page 12, Line 280-288).

7. A paragraph of practical applications for jumpers and sprinter might be suitable. How they should use caffeine during their training. Before all sessions? 48 h after a high-intensity session?

Response: Thank you four this suggestion. A new paragraph has been added to address practical applications (Page 19, line 447-451) “Our results has considerable practical relevance as they are indicating that caffeine can optimize jumping performance in well-trained athletes even when a certain degree of muscle damage and DOMS are present. Thus, ingestion of anhydrous caffeine in some jump training sessions may be an interesting strategy to improve training quality and performance of these athletes”.

8. Line 397. “sprinters” is twice.

Response: This typo error has been fixed accordingly. The text now reads “(sprinters and jumpers)”. (Page 20, line 464).

---

## [Decision Letter · Decision Letter 1]

23 Oct 2019

Effect of caffeine on neuromuscular function following eccentric-based exercise

PONE-D-19-16976R1

Dear Dr. Lima-Silva,

We are pleased to inform you that your manuscript has been judged scientifically suitable for publication and will be formally accepted for publication once it complies with all outstanding technical requirements.

With kind regards,

Daniel Boullosa

Academic Editor

PLOS ONE

Additional Editor Comments (optional):

Reviewers' comments:

Reviewer's Responses to Questions

**Comments to the Author**

1. If the authors have adequately addressed your comments raised in a previous round of review and you feel that this manuscript is now acceptable for publication, you may indicate that here to bypass the “Comments to the Author” section, enter your conflict of interest statement in the “Confidential to Editor” section, and submit your "Accept" recommendation.

Reviewer #1: All comments have been addressed

Reviewer #2: All comments have been addressed

2. Is the manuscript technically sound, and do the data support the conclusions?

Reviewer #1: Yes

Reviewer #2: Yes

3. Has the statistical analysis been performed appropriately and rigorously? 

Reviewer #1: Yes

Reviewer #2: Yes

4. Have the authors made all data underlying the findings in their manuscript fully available?

Reviewer #1: Yes

Reviewer #2: No

5. Is the manuscript presented in an intelligible fashion and written in standard English?

Reviewer #1: Yes

Reviewer #2: Yes

6. Review Comments to the Author

Reviewer #1: The authors have done a good job of completing the manuscript with the proposed comments. Well done!

Reviewer #2: The authors have done a commendable work in improving the manuscript. In this new version, they have changed the focus of their investigation as suggested (from “muscle damage” to muscle performance after eccentric-based exercise), have discussed variables not included in the previous versions (i.e., jumps, CK concentrations) and have included limitations according to the weakness of their experimental design.

I believe that the manuscript is ready for publication.

7. PLOS authors have the option to publish the peer review history of their article (what does this mean?). If published, this will include your full peer review and any attached files.

Reviewer #1: Yes: González-Hernández Jorge Miguel

Reviewer #2: No

---

## [Editor Report · Acceptance letter]

28 Oct 2019

PONE-D-19-16976R1 

Effect of caffeine on neuromuscular function following eccentric-based exercise 

Dear Dr. Lima-Silva:

I am pleased to inform you that your manuscript has been deemed suitable for publication in PLOS ONE. Congratulations! Your manuscript is now with our production department. 

With kind regards,

on behalf of

Dr. Daniel Boullosa 

Academic Editor

PLOS ONE